# Learning to cope with adversarial attacks

Xian Yeow Lee [1]   Aaron Havens [2]   Girish Chowdhary [3]   Soumik Sarkar [1]

## Abstract

The security of Deep Reinforcement Learning (Deep RL) algorithms deployed in real life applications are of a primary concern. In particular, the robustness of RL agents in cyber-physical systems against adversarial attacks are especially vital since the cost of a malevolent intrusions can be extremely high. Studies have shown Deep Neural Networks (DNN), which forms the core decision-making unit in most modern RL algorithms, are easily subjected to adversarial attacks. Hence, it is imperative that RL agents deployed in real-life applications have the capability to detect and mitigate adversarial attacks in an online fashion. An example of such a framework is the Meta-Learned Advantage Hierarchy (MLAH) agent that utilizes a meta-learning framework to learn policies robustly online. Since the mechanism of this framework are still not fully explored, we conducted multiple experiments to better understand the framework's capabilities and limitations. Our results shows that the MLAH agent exhibits interesting coping behaviours when subjected to different adversarial attacks to maintain a nominal reward. Additionally, the framework exhibits a hierarchical coping capability, based on the adaptability of the Master policy and sub-policies themselves. From empirical results, we also observed that as the interval of adversarial attacks increase, the MLAH agent can maintain a higher distribution of rewards, though at the cost of higher instabilities.

[1]Department of Mechanical Engineering, Iowa State University, Ames, Iowa, USA [2]Department of Aerospace Engineering, University of Illinois at Urbana-Champaign, Champaign, Illinois, USA [3]Department of Agricultural and Biological Engineering, University of Illinois at Urbana-Champaign, Champaign, Illinois, USA. Correspondence to: Soumik Sarkar <soumiks@iastate.edu>.

*Proceedings of the 36[th] International Conference on Machine Learning*, Long Beach, California, PMLR 97, 2019. Copyright 2019 by the author(s).

## 1. Introduction

Rapid development of deep neural networks in recent years have sparked subsequent advancements in the field of reinforcement learning. Using deep neural networks as function approximators/policies, the field of Deep Reinforcement Learning (Deep RL) has seen numerous success stories. Examples of Deep RL agents beating Atari (Mnih et al., 2015), learning generalizable policies for robotic manipulation (Ebert et al., 2018) and searching for good neural network architectures (Zoph & Le, 2016) are a few of the examples. As a result, that has subsequently led to Deep RL frameworks getting deployed in real world applications such as health care (Peng et al., 2018; Komorowski et al., 2018), finance (Deng et al., 2017), engineering (Lee et al., 2018; Neftci & Averbeck, 2002) and many more. While the application of Deep RL has been successful in many fields, the security of such systems are also increasingly being scrutinized as an increasing number of studies show that these systems are also not fully reliable.

In the field of deep learning, (Goodfellow et al., 2015) has shown that deep neural networks are extremely fragile and image classifiers constructed from convolutional neural networks (CNN) can be easily deceived to classify images wrongly just by changing the values of a few pixels using the Fast Gradient Sign Method (FGSM). This opens up a large and realistic possibility that systems deployed with deep neural networks may be subjected to adversarial attacks. In the context of real world applications such as detection of dangerous objects in cargo shipments (Jaccard et al., 2016) and autonomous navigation in self-driving vehicles (Rausch et al., 2017), the consequences of misclassifying an object due to a potential adversarial attack are extremely high.

Similarly, the possibility of an adversarial attack on Deep RL systems has also been investigated. In a similar manner of adversarial attacks on image classifiers, (Huang et al., 2017) showed that it is possible to augment the RL agent's observation to fool a trained RL agent to take a sub-optimal action. In addition to that, (Tretschk et al., 2018) demonstrated that instead of fooling the agent to take sub-optimal actions, it is also possible to trick the RL agent into pursuing an adversarially defined goal.

Therefore, it is crucial for an RL agent to be able defend against such adversarial attacks by detecting the presence

of adversaries and mitigate any unwanted consequences. One possible crude defense strategy will be for the agent to raise a warning or autonomously shut itself down in the event of a detected adversary. However, a more elegant defense strategy is for the RL agent to be able to detect and cope with such adversaries in an online manner. A previous study by (Havens et al., 2018) has shown that it is possible for an RL agent to robustly learn policies while subjected adversarial attacks by using a hierarchical meta-learning framework. From a high level point of view, the framework utilizes a master policy that detects whether an adversary is present through the advantages of sub-policies that are optimized for different tasks and decides which sub-policy to use. Nonetheless, the capabilities of the proposed framework are still not fully understood. For example, how exactly is the agent coping when an adversary is detected? Or how does the frequency of attacks affect the performance of the agent?

Inspired by such questions, we perform several experiments to gain insights and a better understanding of the capability of this proposed framework. In this workshop paper, we present the results of our findings in hopes of encouraging future research in the direction of discovering effective and viable defensive strategies.

## 2. Related Works

Multiple studies have shown that RL agent are easily susceptible to adversarial attacks. (Huang et al., 2017) showed that by extending the framework of FGSM to RL agents, the RL agents can be tricked into behaving sub-optimally. (Behzadan & Munir, 2017a) experimented with transferrability of attacks on DQN agents and showed that a properly crafted attack can easily be transferred to another agent with a different model while retaining similar effectiveness. Additionally, more sophisticated adversarial techniques have been proposed by (Lin et al., 2017a). In their experiment, the authors suggests that rather than perturbing the observation of the RL agent repeatedly, it is sufficient to attack the RL agent at strategic time points when the relative preference of the optimal action over the least optimal action is higher than a certain threshold. In addition, the authors also proposed another possible adversarial strategy called the enchanting attack. In this strategy, a series of perturbations are crafted such that the succession of adversarial states will lead the agent to a specific target adversarial state.

In response to the security concern of these Deep RL frameworks, multiple defensive strategies against such adversarial attacks have been proposed. (Behzadan & Munir, 2017b) first showed that a nominally trained RL agents inherently have the capability of recovering from sparse or non-contiguous attack. Their experiment results also demonstrated that RL agents trained under adversarial conditions

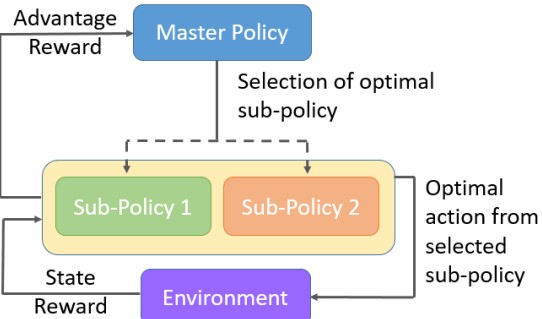

Figure 1. Illustration of the MLAH framework. The Master policy observes the advantages of each sub-policy and decides the optimal sub-policy to employ. The selected sub-policy then acts on the observation from the environment. Note that both the Master policy and selected sub-policy receives the same reward signal from the environment.

turned out to be more robust against test time attacks, thus making adversarial training a good candidate method to robustify RL agents. From a totally different standpoint, (Lin et al., 2017b) approached the problem by comparing the action of the agent acting on the current observation versus the action of the agent acting on a predicted current observation. The predicted current observation is conditioned on previous observations using a future frame prediction model. The authors proposed this framework as a method to detect the presence of an adversary and to use the predicted observation as the surrogate observation when adversaries are detected.

Furthermore, (Havens et al., 2018) proposed an algorithm that detects the presence of adversaries by observing the advantages of sub-policies using a hierarchical framework. Using the proposed algorithm, the results suggested that the learned bias of the RL agent is greatly reduced under adversarial conditions and a robust policy can be learnt while in the presence of unknown adversaries. Leveraging this existing framework, we further explore the viability of using it as a defensive framework.

## 3. Background

As the foundation of the algorithm, we begin with a brief overview of the metalearning shared hierarchies (MLSH) framework as proposed by (Frans et al., 2017). In MLSH, a Master policy parameterized by $\theta$, is tasked to choose a set of sub-policies, each parameterized by $\phi$, to solve a distribution of tasks. The experimental results from the paper shows that a general set of primitive sub-policies can be learned using these framework that can be shared across different tasks. Using these sub-policies, only $\theta$

from the Master policy needs to be re-trained for a given new task as it adaptively chooses the correct sub-policies to solve the new task. The re-training of the Master policy is required due to the non-stationarity of the Markov Decision Process (MDP) introduced by task switching, which may not be known to the agent. The general problem can be stated as the following: Given a set of MDPs $\mathcal{M} : \{m_i\}_n$, where $m_i$ is represented by the tuple $(\mathcal{S}, \mathcal{A}, \mathcal{P}_i, \mathcal{R}_i)$, find a policy to maximize the sum of rewards under the set of tasks: $\sum_t \mathbb{E}_{\sim \rho_0, m_0}[r(s_t)|m_i \in \mathcal{M}]$. $\rho_0$ is an initial state distribution and $m_0$ is a initial MDP distribution. Note that we assume the set of MDP's shares the same state-action space, but may differ in in reward function $\mathcal{R}$ or transition probability $\mathcal{P}$ (specifically due to the adversarial setting).

In a similar fashion, MLAH (Havens et al., 2018) introduces a learned hierarchy where the parameterized Master policy is additionally conditioned on the expectations of each sub-policy. This allows the master agent to detect which task is present in an unsupervised fashion via the reward signal with respect to the expected reward (value or Q-function) of each sub-policy. Depending on $\mathcal{M}$, each sub-policy may specialize in their respective tasks if it is optimal to do so. As each sub-policy improves their performance and value expectation in a task, the master agent improves in selecting the correct sub-policy. An illustrative figure of this framework is shown in Figure 1.

# 4. Methods

The following section describes the method we implemented to demonstrate the coping behaviors of the MLAH framework.

## 4.1. Environment

To demonstrate the coping behaviours that emerged from this framework, we implemented the training of the agent on a custom 2D grid world environment with a similar structure as OpenAI's gym environment (Brockman et al., 2016). In this environment, the goal of the agent is to reach the center coordinates of the grid world. At any time step $t$, the agent is allowed to move up, down, left, right or take no actions. At each time step, the agent gets a reward signal that is equivalent to the scaled temporal difference of the Euclidean distance to the goal, denoted as $|d|$ in Equation 1, between the current time step $t$ and the previous time step $t - 1$. Additionally, the agent is also penalized at each time step with a value of -1 to encourage the agent to reach the goal faster. If the agent reaches the goal, it receives a reward of 100. Last but not least, we set the maximum episode length of each episode to a 100 steps.

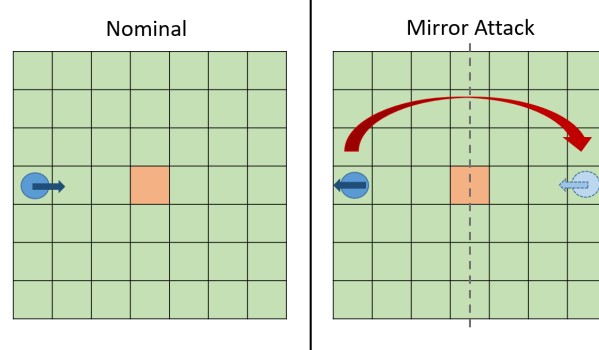

*Figure 2.* Illustration of a symmetric mirror attack on the RL agent about the center vertical axis. Under this attack, the optimal policy changes and the resulting action isn't just sub-optimal but is instead directly leading the agent away from the goal.

$$r(s_t, a_t) = \begin{cases} 100, & \text{if goal is reached} \\ -1 + 10 \times (|d|_t - |d|_{t-1}), & \text{otherwise.} \end{cases} \quad (1)$$

## 4.2. RL Agent

As the grid world environment is relatively low dimensional, we implemented a simple deep neural network architecture for the MLAH agent. We parameterized the the Master policy using 2 dense layers with 16 hidden units on each layer, with a final output dimension of 2, corresponding to two sub-policies for nominal/adversary conditions. Each sub-policy consists of another separate network with 2 dense layers and 32 hidden units each. The sub-policies have output dimensions of 5, representing the actual actions space of the agent in the Grid World environment. Every dense layer in the network is also followed by a $tanh$ activation layer.

## 4.3. Adversarial Functions

In terms of adversarial attacks, we define a few classes of adversarial functions specific to the grid world to deploy on the trained agent.

### 4.3.1. BIAS ATTACK

One class of adversarial function that we implemented was the bias attack. The bias attacks takes in the $(x, y)$ coordinates of the agent and adds a certain value to the coordinates in either the $x$-dimension, $y$-dimension or both. Due to the symmetrical properties of the environment and depending on the location of the agent, a naive bias attack wouldn't affect the nominal policy too much. For example, if the agent in the left half region of the goal gets a bias attack of

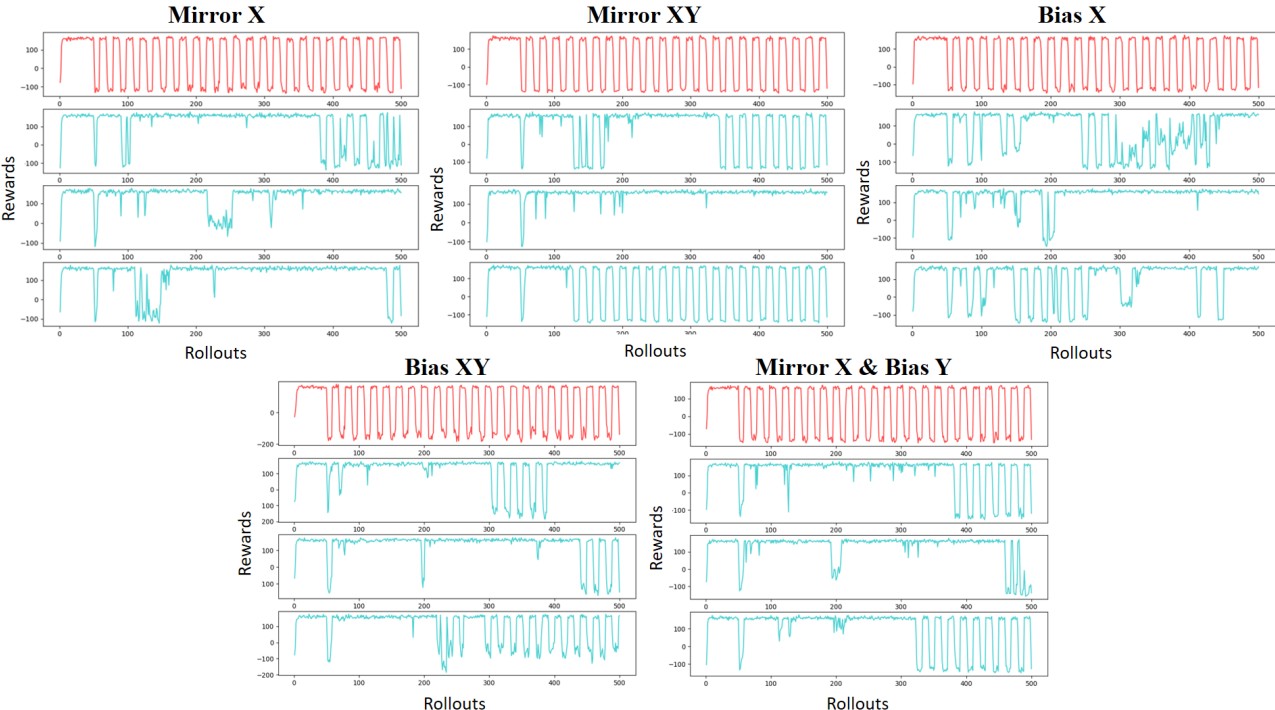

*Figure 3.* Comparison of a nominal agent with just one policy with the MLAH agent across multiple adversary attacks. The performance of the nominal agent are shown in red and the rewards clearly show a periodic presence of adversarial attacks. Performance of the MLAH (across different random seeds) are shown in cyan and there is a clear trend that the MLAH agent is able to cope against the adversarial attacks to maintain a nominal reward.

a certain $\epsilon$ that results in it still being in the left half region, then the optimal action will still be to move right, regardless of whether it is attacked. Hence, we only consider bias attacks that induces a bias of more than half the dimension of the environment to ensure that the optimal policy induced by adversarial observation changes.

### 4.3.2. MIRROR ATTACK

Another class of adversarial function that we implemented was the mirror attack. When the attack is active, the adversary takes in the $(x, y)$ coordinates of the agent and mirrors the coordinate about the center axis of the grid world. In the context of the grid world, this class of attacks has a severe effect on the agent. As illustrated in Figure 2, when the agent is in the region to the left side of the goal, the agent's optimal policy is the move right towards the goal. However, when the attack is applied, the agent is fooled into believing that it is in the region on the right side of the goal. Therefore, based on the adversarially modified observation, the agent's optimal policy is to move left towards the goal, which actually brings it further way from the goal. Similarly, the mirror attack perturbation can be performed across the center $x$-axis, $y$-axis or both.

### 4.3.3. OTHER CLASSES OF ATTACKS

Other classes of adversarial attacks includes composite attacks which can be thought of combinations of the two classes of attacks described above such as mirror-bias attacks. Since this paper serves the to illustrate the emergent coping behaviours shown by the framework in the context of a grid-world, we did not extend the attacks to more sophisticated techniques. However, in environments with higher dimensions, more advanced adversarial functions such as gradient based perturbations such (Goodfellow et al., 2015; Kurakin et al., 2016a;b) may be implemented.

## 5. Results & Discussion

In this section, we present the empirical results and observations of training the RL agent using the MLAH framework in the presence of different adversarial function attacks. We train a nominal RL agent on the custom Grid World with a dimension of 21 x 21 grids with the goal of reaching the center coordinate at $(11, 11)$ with a episodic limit of 100 steps per episode. In each experiment, we first pre-train one of the sub-policies under nominal conditions for 40 roll-outs to ensure that the agent has learn the nominal policy well. After 40 roll-outs, we periodically attack the RL agent with the adversarial attacks defined in the previous section

and jointly train both sub-policies and the master policy for another 450 roll-outs with each roll-out being capped at a 1000 total steps.

## 5.1. Emergent Coping Behaviour of MLAH Against Different Attacks

In Figure 3, we compare the performance of a nominal agent with access to only one sub-policy (shown in red) with the performance of the MLAH agent (shown in cyan with different random seeds) under adversarial attacks across different adversarial functions. In each experiment, we set the adversarial attack to be active under a periodic interval of 10,000 steps. As observed in the first sub-plot of each graph, the agent is able to reach the goal under nominal conditions in less than 10 iterations and consistently receives nominal rewards. However, under periodical adversarial attacks (which begins after 40 iterations), the performance of the agent drops drastically and trend of the rewards clearly reflects the periodic presence of adversarial attacks in all cases.

Conversely, under the same adversarial conditions, the MLAH agent seems to be demonstrating a certain coping behaviour that results in it being able to achieve a high nominal reward even under adversarial attacks. However, we observed that the coping behaviour can sometimes be unstable. When this occurs, the trend of the rewards starts exhibiting the periodic presence of the adversarial attacks. Nonetheless, the MLAH agent has the capability to recover from such instabilities after the Master policy has observed sufficient transitions with poor rewards.

To further understand the coping behaviour exhibited by the MLAH agent, we visualized the actions of the agent in the grid world that was stable under adversarial attacks. Figure 4 illustrates the behaviour of a nominal agent and the MLAH agent under a adversarial mirror attack around the y-axis. As expected, once the Master agent has detected the presence of an adversary, it learns to select a sub-policy that has learnt the inverse of the adversarial state-action mapping. As a result, the agent takes an action that is contrary to the adversarial state, which resolves to the optimal policy under nominal conditions. Similar coping behaviours are also exhibited by the MLAH agent under different adversarial attacks.

## 5.2. Effect of Attack Frequencies on Coping Behaviour

Next, we perform a study on the effect of reducing adversary attack time intervals on the performance of the MLAH agent. Since the Master's selection of the sub-policies depends strongly on it's observation of each sub-policy's advantage, we conjecture that were will be a limit on the ability of the Master agent to detect the presence of an adversary and to switch the sub-policies in the event of sparse adver-

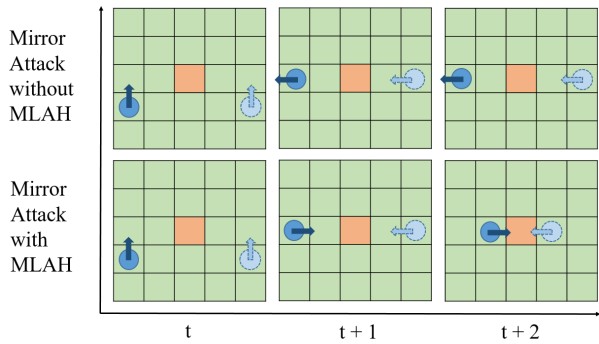

*Figure 4.* Illustration of MLAH agent's coping behaviour under symmetrical mirror attack about the $y$-axis. The MLAH agent learns to use a different sub-policy that maps the adversarial observation to an optimal action that leads it to the goal.

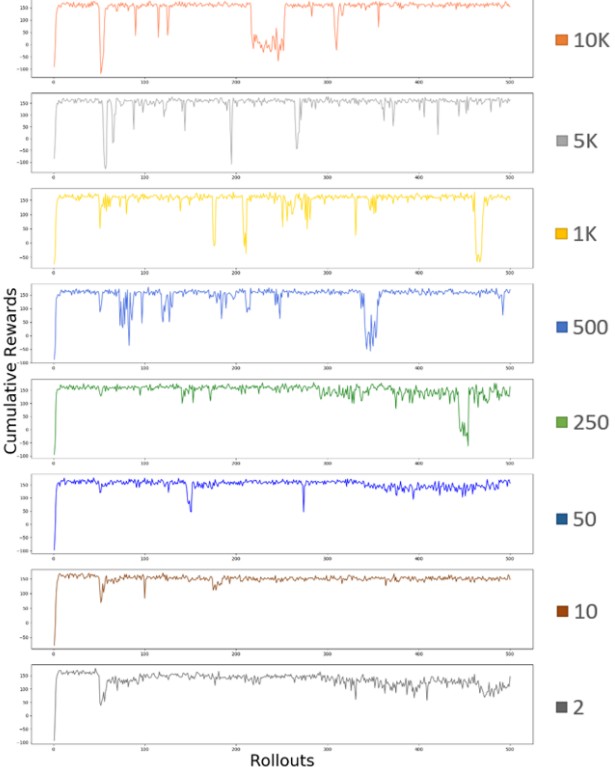

*Figure 5.* Cumulative reward plots of MLAH agent subjected to different intervals of adversarial mirror attacks. A noticeable trend is that as the intervals get smaller, the agent becomes more stable, though at a cost of a lower distribution of rewards with greater variance, as shown in Figure 6.

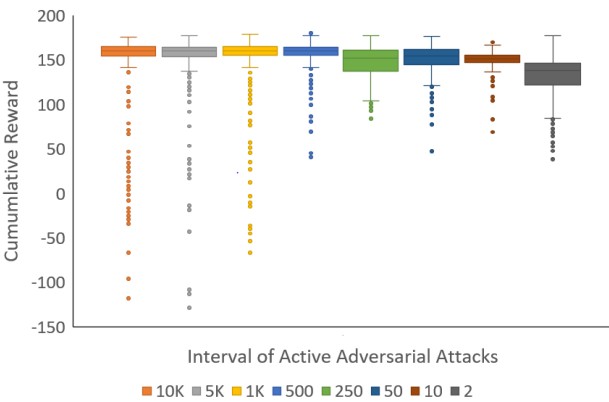

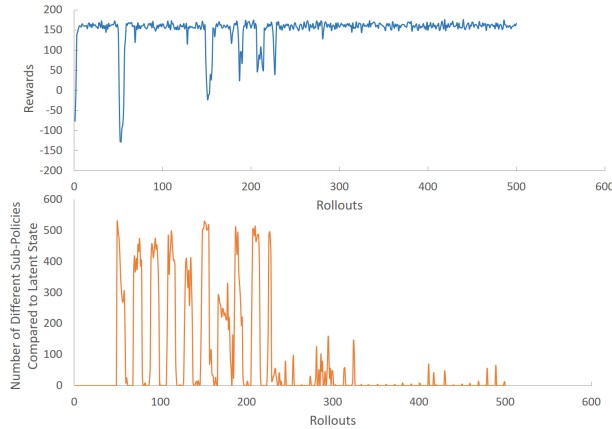

*Figure 6.* Distribution of cumulative rewards for the MLAH agent subjected to adversarial mirror attacks with different intervals. Under long intervals of attacks, the MLAH agent has a higher distribution of rewards, albeit with several outlying points attributed to Master agent's instability. As attack intervals decrease, there are fewer instabilities as evident by fewer outliers, although the distribution of rewards shifts lower with an increase in variance.

*Figure 7.* The plot at the top shows the performance of the MLAH agent under Bias XY attacks with an active adversarial interval of 10000 steps. In the bottom plot, we show the number of times a wrong sub-policy was selected based on an arbitrary definition of 2 latent states (nominal or adversary). We discover that the roles of each sub-policy are not fixed to a particular latent state but can instead evolve to maintain nominal rewards.

sarial attacks. Using the Mirror-$X$ adversarial function as an example, we run multiple experiments on the MLAH agent with different intervals of active adversary attacks starting from a periodic attack at every 10,000 time steps to a periodic attack of every 2 time steps.

Figure 5 shows the performance of the MLAH across the different active adversarial intervals. An intriguing observation is that as the attack intervals decrease, the agent's performance becomes more stable as there are fewer apparent dips in the cumulative reward of the agent across different roll-outs. However, when the the distributions of the cumulative rewards are visualized, as shown in Figure 6, another interesting trend can be observed. Under transient adversarial conditions where the attacks intervals are longer, it can be seen that the majority of the rewards are clustered more tightly together with a median reward greater than 150. However, as the interval of attacks decreases, there is a clear shift in the distribution of the rewards to lower values once the intervals are smaller than 250 steps.

It is also worth nothing that although the reward distributions at longer attack intervals have many outliers, these points can be attributed as the rewards when the Master agent is unstable and fails to select the right policy. Nevertheless, if the outlying points are disregarded, the distribution of rewards are much higher with a smaller variance. In comparison, when the attack intervals are smaller, the distribution of reward have a greater variance and lower median, though with far fewer outlying points.

This gives us an important insight into the effectiveness of MLAH in mitigating adversarial attacks. Given that the adversarial attacks are not too frequent, implementing the

MLAH agent under adversarial attacks can actually result in a higher nominal reward that reflects a much more optimal behaviour. This is because the MLAH agent essentially learns to assign a different sub-policy to a different state-action mapping rather than using the same policy to relearn a different state-action map.

On the other hand, though the nominal agent with only one sub-policy has a lower distribution of rewards, the degradation in performance isn't too extreme either. This can be attributed to two possible causes. First, we believe that given a high frequency of attack, the Master agent isn't able resolve the difference in the underlying state-action mapping. Hence, instead of assigning two sub-policies to two distinct task (nominal and adversary), the Master agent instead learns to simultaneously optimize both sub-policies to seemingly one task. The second possible cause is due to the environment dynamics. In the Grid World, the penalty of the agent for taking an extra step is small relative to the reward of reaching the goal. Hence, in the grand scheme of things, getting deceived and taking a few additional steps may seem insignificant. However, in environments where there are critical states, taking one wrong action that results in a high penalty will definitely result in a lower cumulative reward.

## 5.3. Adaptive Coping Behaviour of Sub-policies

Another intriguing behaviour that we noticed from our experiments was the adaptability of the sub-policies when the Master policy switches the roles of the sub-policies. An anecdotal example of this is shown in Figure 7. Since the

role of each sub-policy was not hard-coded (ie: sub-policy 1 for nominal conditions, sub-policy 2 for adversarial conditions), the decision of selecting a sub-policy for each task was left to the Master policy. The first plot in Figure 7 shows the performance of the MLAH agent under the Bias-XY attack.

To study the behaviour of each sub-policy, we arbitrarily define the nominal condition as the first latent state and the adversarial condition as a second latent state. Next, we plot the number of times a wrong sub-policy was selected with respect to the latent conditions in the second plot of Figure 7. Hence, in an ideal situation where the Master policy consistently selects the correct sub-policy for the correct latent condition, the second plot in Figure 7 should be a horizontal line at zero. However, we observed that the Master policy initially selects a high ratio of the wrong sub-policy before leveling out to 0 after the 300th roll-out. Nonetheless, the cumulative rewards in the initial phase remains largely nominal in the top plot.

This signifies that although the Master's initial selection of sub-policy was wrong by our arbitrary definition, the sub-policies themselves can adapt to cope with the adversarial attacks to maintain nominal rewards. In the latter phase of the roll-outs, the Master policy starts to select the correct sub-policy according to our definition and once again, the sub-policies switch roles to adapt to the different state-action mapping. One important implication of this characteristic is that this framework can potentially be used to cope against evolving or multiple strategies of adversarial attacks since the sub-policies themselves have the capacity to evolve and adapt. However, if more than one distinct attack strategies are being applied in a short interval of time, the sub-policies might not be able to adapt as quickly and a third sub-policy might be required to mitigate the attacks.

## 6. Conclusion & Future Works

In this work, we demonstrated the coping behaviours of a MLAH agent that was subjected to multiple different adversarial attacks in a Grid World environment. We observed that the Master policy is capable of a selecting a sub-policy that learns to map an adversarial observation to action that leads to nominal rewards. Furthermore, we perform a study on the effect of attack intervals on the agent's performance. We find that for longer intervals of attacks, the MLAH agent is able to distinguish between different underlying task distributions to select the right sub-policy to cope. Nevertheless, the process can sometimes be unstable. In contrast, for shorter intervals of attacks, the switching of the sub-policies becomes more stable but at the cost of a lower distribution of rewards and a greater variance.

Additionally, we also find that the sub-policies themselves

can be adaptive when the Master policy fails to select the sub-policy, hence adding a hierarchy of robustness to this framework. Future works include modifying the algorithm to stabilize the selection process of the sub-policies. Another interesting avenue of work is to look at the effectiveness of this framework when the agent's action signals and reward signals are corrupted.

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
