# OpenReview forum: "Learning to Cope with Adversarial Attacks"
_ICML.cc/2019/Workshop/RL4RealLife — Submitted to RL4RealLife 2019_

### Official Review · AnonReviewer1 · 2019-05-21
**Empirical Evaluation of MLAH**

**Rating:** 2
**Confidence:** 4

**Review:**

This paper presents an empirical evaluation of Meta-Learned Advantage Hierarchy (MLAH). MLAH is a deep RL algorithm that is trained to be robust to adversarial attacks, as it meta-learns to switch policies under attack. The paper demonstrates some results of MLAH on a gridworld task.

The main issue with the paper is that it does not appear to add much over the original MLAH paper. The paper states that the framework was not fully explored in the original paper, and so they do more experiments to demonstrate its coping behavior. The original paper had experiments comparing MLAH with a vanilla agent on inverted pendulum and this same gridworld, and it had plots showing MLAH's policy selection. The contributions of this paper seem to be 1) showing explicitly that the adversarial policies do the right thing; and 2) new experiments showing how MLAH handles different frequencies of adversarial attack.

All the experiments in the paper are performed on a grid world - this doesn't seem to be very real life to me.

Since the paper is entirely about empirical evaluation of MLAH, it should really have a more precise description of MLAH, instead there is a single paragraph describing MLAH at a high level.

Clarity:
The paper is mostly clear, but needs a clearer more precise description of the MLAH algorithm being evaluated.

Originality:
The paper does not present any new algorithm or application, and most of the experiments are repeated from the MLAH paper. The only original parts are 2 new experimental results.

Significance:
The paper does not seem to provide any new signficant results or understanding over the original MLAH paper.

Pros:
- Adversarial robustness is a potential challenge of applying RL to real life problems

Cons:
- Very little new contribution over existing published MLAH paper.
- Grid world is not a real life problem

---

### Official Review · AnonReviewer2 · 2019-05-23
**Application of a previously published learning approach on a (slightly too) simple task**

**Rating:** 2
**Confidence:** 4

**Review:**

The paper studies the effectiveness of the "Meta-Learned Advantage Hierarchy" with respect to a number of attacks that come from sudden environment changes. The study is conducted on a synthetic Grid Path environment for two types attacks and their combinations. Through several experiments the paper demonstrates the robustness of the MLAH approach for those attacks and studies the sensitivity of the method related to the frequency of attacks. The paper is quite easy to read and follow.

In my opinion, in the context of the workshop goals, the paper does not present substantial enough contributions due to the simplicity of the toy environment and attacks used for the experiments. Ideally, the experiments should be conducted on a more difficult task, which highlights the effectiveness of the MLAH framework to realistic malicious mutations of the environment more likely to happen in non-simulated tasks. On the difficulty of the task, without switching to a different one, authors could vary the grid size during training (increase up to the point where learning is no longer possible), add obstacles which make moving in the environment more difficult or consider stochasticity (unlike just pre-programmed "attacks" with a certain frequency).

The paper would also benefit from more clarity around the following:
- describe observations which go at each decision step to the agent
- describe the learning algorithm used for training the nominal and adversarial policies
- motivate the structure of the reward given with context around the fixed scaling factor

In Figure 5. I expect the first 40 roll-outs to represent the nominal policy being trained on the task. At roughly the same time for all intervals when the adversarial attacks start, they seem to have strongly different impact across the plots, but I would expect them to be identical - is this simply due to variability in difficulty of the attack?

On the usage of term "adversarial": perhaps "malicious" would be a better term than adversarial:
- adversarial attacks are commonly thought of as changes in the observations space through the differentiable loss (like FGSM)
- they are typically influenced by the agent's state, while here what is referred to as "adversarial functions" are changes to the environment regardless of how well the agent performs (are basically sudden environment changes).

---

### Decision · Program_Chairs · 2019-05-28

Reject